# Specialised orthotic care to improve functioning in adults with neuromuscular disorders: protocol of a prospective randomised open-label blinded end-point study

Elza van Duijnhoven  ,[1] Fieke Sophia Koopman,[1] Jana Antonius Maria Tuijtelaars,[1] Viola Altmann,[2] Rimke Lagrand,[1] Johanna Maria van Dongen,[3] Frans Nollet,[1] Merel-Anne Brehm[1]

¹Department of Rehabilitation Medicine, Amsterdam Movement Sciences, Amsterdam UMC, University of Amsterdam, Amsterdam, The Netherlands
²Klimmendaal Rehabilitation Center, Arnhem, The Netherlands
³Department of Health Sciences, Faculty of Science, Vrije Universiteit Amsterdam, Amsterdam, The Netherlands

**Correspondence to**
Elza van Duijnhoven;
e.vanduijnhoven@amsterdamumc.nl

## ABSTRACT

**Introduction** People suffering from leg muscle weakness caused by neuromuscular disorders (NMDs) are often provided with leg orthoses to reduce walking problems such as increased walking effort, diminished walking speed, reduced balance and falls. However, evidence for the effectiveness of leg orthoses to improve walking in this patient group is limited and there is an absence of standardised practice in orthotic prescription. In 2012 a Dutch multidisciplinary guideline was developed aimed to standardise the orthotic treatment process in NMD. Although application of the guideline in expert centres (specialised orthotic care) seems beneficial regarding clinical effectiveness, larger studies are necessary to confirm results and investigate cost-effectiveness. Therefore, this study aims to examine the effectiveness and cost-effectiveness of specialised orthotic care compared with usual orthotic care in adults with slowly progressive NMD.

**Methods and analysis** A prospective randomised open-label blinded end-point study will be performed, in which 70 adults with slowly progressive NMD are randomly assigned to specialised orthotic care (intervention) or usual orthotic care (control). Outcome measures are assessed at baseline and at 3 and 6 months follow-up. The primary endpoints are gross walking energy cost (J/kg/m) assessed during a 6 min walk test and achievement of personal goals, measured with the Goal Attainment Scale. Secondary endpoints include walking speed, gait biomechanics, stability, physical functioning, falls and fear of falling, perceived fatigue and satisfaction. For the economic evaluation, societal costs and health-related quality of life will be assessed using cost questionnaires and the 5-Level version of EuroQol 5 Dimension, retrospectively.

**Ethics and dissemination** The study is registered in the Dutch trial register (NL 7511) and the protocol has been approved by the Medical Ethics Committee of the Academic Medical Center in Amsterdam. Results will be presented at national and international scientific conferences and disseminated through peer-reviewed journals and media aimed at a broad audience including patients.

---

### Strengths and limitations of this study

► A broad range of outcome measures, both objective and self-reported, will be collected, resulting in a unique dataset of both functional effects, underlying biomechanical working mechanisms and costs.

► The two primary outcome measures to evaluate functional effects include walking energy cost and achievement of personal goals measured with the Goal Attainment Scale, which is chosen as an additional primary outcome measure to capture the diversity of orthotic goals relevant to the individual.

► Evidence will be obtained in a large group of adults with different slowly progressive neuromuscular disorders, which increases the generalisability of results.

► An economic evaluation will be performed alongside the study to provide insights into the cost-effectiveness of specialised orthotic care versus usual orthotic care.

► To account for the expected variability in treatment components of usual orthotic care, which is a possible limitation that could complicate the identification of underlying mechanisms of action between treatments, the process of usual orthotic care will be extensively documented.

---

## INTRODUCTION

People with slowly progressive neuromuscular disorders (NMDs), such as postpolio syndrome and Charcot Marie Tooth disease, frequently suffer from leg muscle weakness.[1] Leg muscle weakness will change the gait pattern,[2] causing walking problems such as increased walking effort,[3 4] diminished walking speed,[3 4] pain,[5] balance problems and falls.[6 7] These walking problems may restrict the patients' performance of daily physical activities[8] and negatively affect their independency and quality of life.[8 9] Due to the

progressive nature of NMD, walking problems will gradually increase over time,[9] which will further negatively affect physical mobility, independency and quality of life.

Leg orthoses are provided to reduce walking problems in people with NMD.[10] A leg orthosis is an 'externally applied medical device encompassing (part of) the leg and foot, used to modify the structural and functional characteristics of the neuromuscular and skeletal systems'.[11] In general, an ankle-foot orthosis (AFO) is provided in case of distal leg muscle weakness, which often includes weakness of the foot dorsiflexors and plantarflexors.[10 12] Accordingly, the AFO should support foot clearance during swing by restricting plantarflexion and compensate for plantarflexor weakness by restricting dorsiflexion during late stance.[10] Knee-ankle-foot orthoses (KAFOs) are provided for (additional) proximal leg muscle weakness, particularly weakness of the quadriceps, to stabilise the knee joint during the stance phase of gait and allow safe weight-bearing.[10 13]

In previous research, various types of AFOs for calf muscle weakness[14] and KAFOs for quadriceps weakness have been found to improve walking in people with NMD.[15] However, the strength of the evidence is rather limited due to a lack of proper study designs, relatively small sample sizes, and heterogeneity in types of leg orthoses and control conditions studied. Accordingly, there is a lack of evidence-based and standardised guidance on the application of leg orthoses. Furthermore, many different types of leg orthoses are being prescribed in current orthotic practice in NMD, varying largely in orthotic properties and effectiveness, with both good and suboptimal treatment outcomes in terms of improving walking.[16 17]

In 2012, a Dutch multidisciplinary guideline for leg orthoses in adults with slowly progressive NMD was developed with the aim to standardise the treatment process and improve treatment outcomes.[18] The guideline was developed according to proposed national and international frameworks[19–21] and compromises the entire process of orthotic treatment in systematically divided steps, as well as treatment algorithms for the selection of leg orthoses. According to the guideline, the care need should be individually characterised in terms of the personal health problems, goals and gait deviations to be addressed, caused by the underlying impairments. Based on the care need, the orthotic goals should be defined and matched with the orthosis design. Gait training after delivery of the orthosis after is essential to maximise effectiveness.[22] Finally, it is important to systematically evaluate the effectiveness as well as user experiences to ensure the functionality of the orthosis.[23]

At this moment, the guideline is not widely applied throughout the Netherlands, as it requires sufficient expertise of the multidisciplinary care team involved and facilities for advanced 3-dimensional (3D)-gait analysis and gait training. However, prescribing leg orthoses according to the guideline (i.e. specialised orthotic care) seems promising in terms of clinical effectiveness as recently shown in two uncontrolled trials.[16 24] In individuals with calf muscle weakness due to NMD, individually stiffness-optimised AFOs have been shown to reduce walking effort to a much greater extent compared with AFOs prescribed in usual orthotic care.[24] Furthermore, in polio survivors,[16] the increment in walking effort with KAFOs prescribed in specialised orthotic care was reduced with 18% towards normative values, when compared with usual care KAFOs. At this moment, the effectiveness of leg orthoses prescribed within specialised orthotic care on functioning needs to be investigated in a larger group of individuals with slowly progressive NMD in comparison to usual orthotic care to strengthen evidence for the possible benefit.

While good-quality studies are needed to strengthen the evidence for the effectiveness of specialised orthotic care in NMD, it is also imperative to investigate whether the treatment is cost-effective, which is currently unknown. Since new and expensive orthotic devices increasingly become available, it is important to assess their associated costs.[25] Also, at this moment, an important problem in usual orthotic care concerns the low compliance of patients wearing their devices,[26] which in turn leads to an inefficient use of healthcare resources.[27] Optimally matching orthoses with the personal needs and goals of the patient, according to the guideline, could lead to gains in clinical effectiveness and subsequently a higher compliance. In this respect, specialised orthotic care could improve the efficiency of (healthcare) resource, but this has not yet been investigated.

The aim of this study is to examine the effectiveness and cost-effectiveness of specialised orthotic care compared with usual orthotic care on functioning in adults with NMD. We hypothesise that specialised orthotic care is more effective in terms of reducing walking effort and achieving personal goals when compared with usual orthotic care. Furthermore, specialised orthotic care is expected to be cost-effective compared with usual orthotic care from a societal and healthcare perspective.

## METHODS AND ANALYSIS
### Study design
The study is designed as a randomised open-label blinded end-point study that is prospectively registered at the Dutch Trial Register under number NL7511. The trial protocol was developed according to the Standard Protocol Items: Recommendations for Interventional Trials guidelines[28] (online supplemental file 1). Recruitment started in March 2019 and is foreseen to end at December 2021. Measurements are conducted at baseline (T1) and 3 and 6 months after orthotic treatment is given (T2 and T3, respectively). To determine the cost-effectiveness, an economic evaluation is performed alongside the study. An overview of the study design is shown in figure 1.

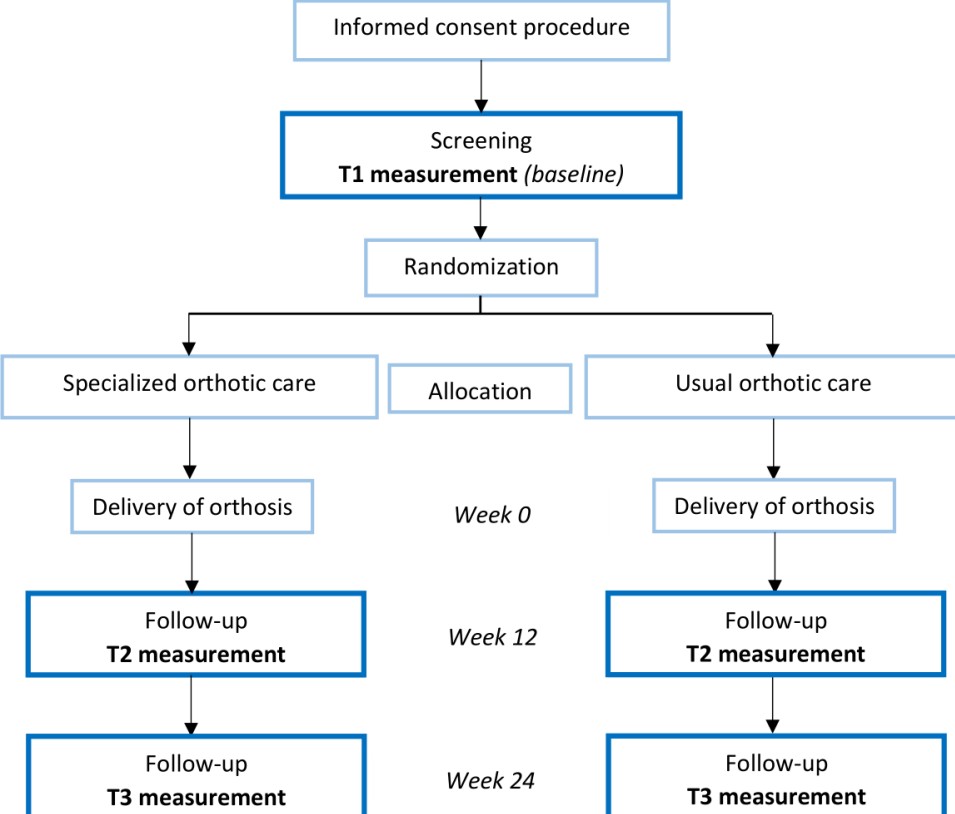

**Figure 1** Overview of the study design.

## Study population

We intend to include 70 adults with slowly progressive NMD (such as Charcot-Marie-Tooth disease, post-polio syndrome, inclusion body myositis or myotonic dystrophy) or peripheral nerve injury who have non-spastic leg muscle weakness. For eligibility, patients may or may not use an orthosis provided in usual care at the time of recruitment. Patients will be recruited from rehabilitation centres and hospitals throughout the Netherlands and through the Dutch Association for Neuromuscular Diseases. If a patient is willing to participate, a screening visit is planned. During the screening visit, written informed consent (online supplemental file 2) is obtained and the investigator, in close collaboration with a rehabilitation physician, will check the inclusion and exclusion criteria.

Inclusion criteria are: (1) minimum age of 18 years, (2) weakness of the calf muscles (ie, Medical Research Council (MRC) scale[29] score <5 or not being able to make a heel-rise on one leg >3 times) and/or weakness of the quadriceps (ie, MRC score <5) (3) experiencing walking problems such as increased walking effort, pain and/or impaired balance during standing and/or walking, (4) able to walk for 6 min at comfortable speed with or without assistive device (eg, cane, crutch, walker), (5) indicated for an orthosis based on physical examination and 3D gait analysis and (6) motivated to use an orthosis. An exclusion criterion is: (1) insufficient mastery of the Dutch language.

## Sample size

The sample size of this study is based on a power analysis of the expected differences in walking effort (defined as gross walking energy cost) and the achievement of personal treatment goals between the intervention and control group. According to previous studies,[16 17] a difference in change in walking energy cost of 0.60 J/kg/m from baseline to 6 months post-treatment is expected between usual orthotic care and specialised orthotic care, where a difference of at least 0.45 J/kg/m is considered as clinical relevant.[3] Based on an intention-to-treat analysis, alpha of 0.05, power of 80% and using an estimated correlation coefficient of the repeated measures of 0.76 and SD of 0.90 J/kg/m, 30 patients per treatment group are necessary. The goal is to include 70 patients allowing for a 15% drop-out. This sample size also allows the detection of 1 point difference in personal achievement goal scores (Based on an intention-to-treat analysis, alpha of 0.05, power of 80%, and using an estimated correlation coefficient of the repeated measures of 0.70).

## Randomisation and blinding

After baseline measurement (T1), participants will be randomly assigned in a 1:1 ratio to receive specialised orthotic care (intervention group) or usual orthotic care (control group). The randomisation scheme will be computer generated in Castor (Castor EDC, Amsterdam, the Netherlands) and uses blocks of random sequences with variable sizes (2, 4 and 6). Patients will be stratified by

disease severity defined as distal leg weakness versus (distal and) proximal leg weakness, respectively. Outcomes will be assessed by a blinded and independent assessor. When patients are informed of group allocation, they will be instructed not to reveal this to the outcome assessor.

## Intervention

### Specialised orthotic care

Patients in the intervention group will receive specialised orthotic care at an expert centre that has implemented the Dutch guideline for leg orthoses[18] and has facilities for 3D gait analysis, gait training and fabricating custom-made leg orthoses. For the orthotic care process, protocols of the guideline will be used and treatment integrity checks will be performed to ensure compliance with the guideline.

First, the health problems and goals of the patient will assessed by means of an interview, questionnaires, physical examination and 3D gait analysis. Based on these assessments, the multidisciplinary care team will identify how the patients' problems and goals are best matched to an orthotic solution. The desired level of functioning for the patient and the required orthotic functions will be described in a care plan. Accordingly, the orthosis will be custom fabricated by an orthotic technician. After delivery of the orthosis, instructions on use and maintenance will be given to the patient and, when indicated, the patient receives gait training supervised by an experienced physiotherapist. Finally, after 3 months of using the orthosis, experiences of the patient as well as the desired level of functioning and orthotic functions will be evaluated. When needed, the orthosis will be adjusted based on the evaluation. The provision process will be extensively documented for each participant individually by an investigator who is not blinded.

### Usual orthotic care

Participants in the control group will receive usual orthotic care from a rehabilitation physician at their regular care centre. Treatment can vary across centres and may include provision of orthopaedic shoes, provision of off-the-shelf or custom-made AFOs or provision of off-the-shelf or custom-made KAFOs according to the discretion of the practitioners and local policy of the centre. For participants who already use an orthosis at baseline, usual orthotic care will concern the provision of a new orthosis which may be identical of different from the orthosis being used. The provision process in the usual care centres will be extensively documented for each participant individually by an investigator who is not blinded.

## Outcome measures

Relevant demographic variables, anthropometrics and clinical characteristics will be measured at baseline (T1). Clinical characteristics to be evaluated include: (1) manually assessed muscle strength of the left and right hip flexors and extensors, hip abductors and adductors, knee flexors and extensors, ankle plantar and dorsal flexors and ankle invertors and evertors, scored according to the MRC scale and (2) passive range of motion of the hip, knee and ankle joints (left and right). Furthermore, (3) occurrence of joint deformities and (4) impairments in sensory function will be evaluated.

Primary outcomes and secondary outcomes (described below) will be assessed at baseline (T1) and at 3 and 6 months after orthotic treatment (T2 and T3, respectively). Additionally, orthotic properties (such as type, weight, material and orthotic functions) and adverse events due to the use of the device (such as pressure sores, pain, muscle soreness) will be documented. The use of any medication during the study will be monitored at all measurement time points and documented in the cost questionnaires. All collected outcomes will be entered into a Castor database. An overview of all outcomes per measurement visit is given in table 1.

## Primary outcomes

Primary outcomes are walking energy cost and the achievement of personal treatment goals. Walking energy cost will be determined during a 6-minute walk test (6MWT) in which participants walk at a self-selected comfortable speed on an indoor oval track in their preferred walking direction (which will be kept similar over measurement time points). Simultaneously, oxygen uptake ($VO_2$) and the respiratory exchange ratio (RER) will be measured breath by breath with the Cosmed K5 portable gas analysis system (Cosmed, Rome, Italy). Mean $VO_2$, RER and walking speed values will be obtained at steady state from the last 3 min of the 6MWT using a custom-written Matlab script (V.2019; MathWorks, Natick, Massachusetts, USA). From these outcomes, walking energy cost in J/kg/m will be calculated by using the following formula: $((4.940 \times RER) + 16.040) \times VO_2 / walking\ speed$ where $VO_2$ is in mL/kg/min.[30] The assessment of walking energy cost has previously been shown to be reliable in patients with NMD.[31 32]

The achievement of personal treatment goals will be quantified with the Goal Attainment Scale (GAS) to capture the diversity of orthotic treatment goals important to the individual patient.[33] At baseline, two goals that are highly relevant to the individual will be determined with the patient based on what they wish to achieve in daily life in terms of activities and participation according to the International Classification of Functioning, Disability and Health (ICF) framework.[20] Subsequently, personal GAS scales will be determined by creating six distinct levels of outcome ranging from −3 to +2, where the desired attainment goal of the patient is defined as 0 and the current situation is defined as −2. Achievement of the goals will be scored at T2 and T3 as −3=worsened, −2=unchanged, −1=somewhat less than expected, 0=expected outcome, +1 = somewhat more than expected and +2 = much more than expected. Improvements of at least two points will be regarded as

**Table 1** Overview of outcomes per measurement session

| | | Baseline | Follow-up | |
| --- | --- | --- | --- | --- |
| | | T1 measurement (screening) | T2 measurement (12 weeks after delivery) | T3 measurement (24 weeks after delivery) |
| **Primary outcomes** | | | | |
| Walking energy cost | 6 MWT | X | X | X |
| Personal goals | GAS | *Setting goals* | X | X |
| **Secondary outcomes** | | | | |
| Walking speed | 6 MWT | X | X | X |
| Gait biomechanics | 3DGA | X* | X | X |
| Stability | NRS | X | X | X |
| Physical functioning | SF36-PF | X | X | X |
| Fear of falling | FES | X | X | X |
| Fall rate | Questionnaire | X | X | X |
| Fatigue | FSS | X | X | X |
| Satisfaction | D-Quest | X† | X | X |
| **Additional outcomes** | | | | |
| Demographics | Intake | X | | |
| Anthropometrics | Physical exam | X | | |
| Muscle strength | Physical exam | X | | |
| Joint passive range of motion | Physical exam | X | | |
| Sensory function | Physical exam | X | | |
| Orthotic properties | CRF | X† | X | X |
| Adverse events | CRF | | X | X |
| **Economic evaluation** | | | | |
| Resource use‡ | Cost questionnaire | X | X | X |
| Health-related quality of life‡ | EQ-5D-5L | X | X | X |

*Gait analysis conditions at baseline that will be used for statistical analysis concern walking with shoes only or walking with the old orthosis (in case a participant uses an orthosis at baseline).
†Outcomes will only be assessed in case a participant uses an orthosis at baseline.
‡Outcomes for the economic evaluation will also be assessed directly after delivery of the orthosis.
CRF, clinical report form; 3DGA, 3-dimensional gait analysis; D-Quest, Dutch version of the Quebec User Evaluation of Satisfaction with Assistive Technology; EQ-5D-5L, 5-Level version of EuroQol 5D; FES, Falls Efficacy Scale; FSS, Fatigue Severity Scale; GAS, Goal Attainment Scale; 6MWT, 6-minute walk test; NRS, Numeric Rating Scale; SF36-PF, Physical Functioning Scale of the Short-Form Health Survey.

clinical relevant.[34] The investigator defining and scoring the GAS, followed a course in applying GAS within a rehabilitation context.[35]

### Secondary outcomes

Secondary outcomes include walking speed (measured during the 6MWT), gait biomechanics (explained below), perceived stability during walking (assessed with a 11-point Numeric Rating Scale (NRS)), perceived physical functioning (PF) (assessed with the Short-Form Health Survey PF scale (SF36-PF),[36] frequency of falls and fear of falling (assessed with the short version of the Falls Efficacy Scale (FES),[37] perceived fatigue (assessed with the Fatigue Severity Scale (FSS)[38] and satisfaction with the orthosis (assessed with the Dutch version of the Quebec User Evaluation of Satisfaction with Assistive Technology (D-QUEST)[39] added with self-designed items).

### Gait biomechanics

Gait biomechanics will be assessed with a 100 Hz 12-camera 3D motion capture system (VICON MX V.1.3) and two adjacent 1000 Hz force plates (OR6-7; AMTI, Watertown, Marssachusetts, USA). Preparations include placement of reflective markers on the body according to the Plug-In Gait model. After a static calibration, participants will be asked to walk along a 12 m long walkway. Conditions that will be assessed include barefoot walking (T1), walking with the current orthosis if applicable (T1) and walking with the new orthosis (T2 and T3). Per condition, at least three valid trials will be used for analysis containing a clear stance phase on the force plate for both feet and full visibility of all markers during the gait cycle. Joint angles, net joint moments and joint powers around the hip, knee and ankle will be calculated per trial and averaged over three trials. Spatiotemporal gait parameters (such as step

length and step width) and relevant kinematic and kinetic variables (such as maximal ankle dorsiflexion angle, peak ankle power, knee angle and knee moment at midstance, hip flexion angle and maximal hip, knee and ankle angle during swing and progression of the centre of pressure) will be obtained from these averaged data and used for analysis.

## Economic evaluation

The economic evaluation will be performed from a societal and a healthcare perspective. When the societal perspective is applied intervention costs (costs directly related to the delivery of the orthosis) healthcare costs (costs related to visits to general practitioners, medical specialists, and/or therapists, medication use and assistive devices) informal care costs, unpaid productivity costs, as well as costs related to productivity losses due to being absent from work (absenteeism) and productivity losses due to reduced productivity while being at work (presenteeism) will be included and assessed with a cost questionnaire. When the healthcare perspective is applied, only costs accruing to the formal Dutch healthcare system will be included. To collect data on resource use, participants are asked to fill in cost questionnaires at baseline (T1), directly after delivery of the orthosis and at 3 and 6 months post orthotic treatment (T2 and T3). All cost categories will be valued in accordance with the Dutch manual for costing studies in healthcare.[40]

Outcome measures used for the economic evaluation will be quality-adjusted life years (QALYs), walking energy cost and GAS scores. QALYs will be based on the 5-Level version of EuroQol 5 Dimension (EQ-5D-5L), administered at each measurement point (T1, T2 and T3). The patients' EQ-5D-5L health states will be converted into utility scores using the Dutch tariff.[41] Subsequently, QALYs will be estimated by multiplying the patients' utility scores by the time spent in a certain health state.

## Statistical analysis

Walking energy cost and secondary outcomes at each measurement point (T1, T2 and T3) will be analysed with linear mixed models for repeated measurements to investigate the effectiveness of specialised orthotic care compared with usual orthotic care over time, adjusted for stratification and differences in baseline scores. Time and study group will be included as dependent variables. The differences in GAS scores between groups at 3 and 6 months follow-up will be tested with non-parametric Mann-Whitney U tests, as GAS scores are ordinal. All comparisons between groups are based on an intention-to-treat analysis.

For the economic evaluation, mean differences in total costs and effects between groups will be estimated using seemingly unrelated regression analyses. To account for the skewed nature of cost data, 95% CIs surrounding cost differences will be estimated using bias corrected and accelerated bootstrapping with 5000 replication. Subsequently, incremental cost-effectiveness ratios will

be calculated by dividing the mean differences in costs by the mean differences in effects. To illustrate the joint uncertainty surrounding costs and effects, cost-effectiveness planes and cost-effectiveness acceptability curves will be plotted. In a cost-effectiveness acceptability curve, the probability of specialised orthotic care being cost-effective compared with usual orthotic care is plotted for a range of willingness to pay values (ie, the amount of money decision-makers are willing to pay per unit of effect gained).

All statistical analyses will be performed using SPSS (V.25; IBM SPSS) and a statistical significance of $p < 0.05$ will be used in this study.

## Patient and public involvement

Patients were actively involved in the preparation of this study through participating in meetings with the research group in the development stage of the study protocol and by providing feedback on the study procedures and the patient information documentation. During the conductance of the study, patients will be informed about the progress and involved in patient recruitment and the interpretation, reporting and dissemination of the results.

## DISCUSSION

The aim of this study is to examine the effectiveness and cost-effectiveness of specialised orthotic care compared with usual orthotic care on functioning in adults with slowly progressive NMD. This study has several strengths.

First, a broad range of outcome measures on consecutive time points before and after orthotic treatment will be collected. For the examination of effectiveness, we will not only analyse objective outcome measures, such as walking energy cost and gait biomechanics, but also patient-reported outcome measures, like satisfaction with the provided orthosis. Besides, personal treatment goals will be set at the activity and participation level of the ICF and are therefore considered to be highly relevant to the patient. By assessing outcome measures from multiple perspectives, we will be able to fully capture the functional effects of specialised orthotic care and, at the same time, gain insight in the underlying biomechanical working mechanisms. Additionally, compared with previous research, evidence will be obtained in a large group of adults suffering from leg muscle weakness, caused by many different slowly progressive NMDs, which increases the generalisability of results.

Second, participants in the control group will be treated in a diverse sample of usual care centres in which leg orthoses are prescribed throughout the Netherlands. This allows for a broad comparison between specialised orthotic care and orthotic care as applied in current practice, which could be of importance for the improvement of overall quality and efficiency of orthotic care. On the other hand, treatment components of the usual care process are expected to differ among centres due to local policies and available facilities. This could complicate the

identification of underlying mechanisms of action that could explain differences in treatment outcomes between groups. To enable cautious exploration of potential mechanisms that might lead to beneficial treatment outcomes, which could be useful for future implications, the process of usual orthotic care will be extensively documented.

Finally, a major strength is the economic evaluation that will be performed alongside the study. Since expensive orthotic devices increasingly become available due to technological developments, it is important to not only assess whether orthotic devices are effective, but also whether their additional health effects are worth their additional costs. This study will be the first to provide insights into the cost-effectiveness of orthotic care in NMD in general and of specialised orthotic care compared with usual orthotic care in particular.

In conclusion, this study aims to examine the effectiveness and cost-effectiveness of specialised orthotic care compared with usual orthotic care in adults with NMD. Insights could lead to improvements in the quality of orthotic care resulting in improvements in overall functionality of adults with NMD in daily living. Consequently, insights could lead to a more efficient use of already scarce (healthcare) resources.

## ETHICS AND DISSEMINATION

The study protocol was approved by the Medical Ethics committee of the Amsterdam UMC, location Academic Medical Center, ABR-number 67268. The study is registered at the Dutch Trial Register (NL7511) and will be performed in accordance with good clinical practice guidelines. An independent monitor of the Amsterdam UMC will monitor the study multiple times during the conduction of the study. Aspects that will be monitored will include inclusion rate, trial master file, informed consent process, inclusion and exclusion criteria, randomisation, trial procedures, source data verification, safety reporting and closing and reporting. The investigator will be encouraged to maintain the blinding as far as possible and code breaks should only occur in exceptional circumstances. Important protocol changes will be recorded (a new protocol version number will be assigned) and reported to the Medical Ethics Committee. When patients sign their informed consent, they will receive a participant ID, which will be coupled to all the data collected. Forms will be stored in a locked cabinet to assure anonymity. Only persons involved in the study have access to these forms. Insurance has been taken out for participation of patients in the study. After completion of the study, positive as well as negative or inconclusive results will be submitted to a peer-reviewed journal and presented at national and international scientific conferences. The study data sets and codes of the data analysis are available on request. Furthermore, results will be disseminated through other media aimed at a broader audience including patients. Participants will be informed about the results of the study by means of a newsletter.

**Contributors** M-AB, FSK and FN conceived the study. M-AB, FSK, FN, JMvD and EvD contributed to the study design and methods. M-AB, FSK, FN, JAMT, RL, VA and EvD participated in logistical planning of the study. EvD wrote the manuscript and is responsible for data acquisition and analysis. All authors conceived, provided feedback and approved the final version of the manuscript.

**Funding** This study was supported by ZonMw, The Netherlands Organisation for Health Research and Development, grant number: 853 001 103.

**Competing interests** None declared.

**Patient consent for publication** Not required.

**Provenance and peer review** Not commissioned; externally peer reviewed.

**ORCID iD**
Elza van Duijnhoven http://orcid.org/0000-0002-7876-046X

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
