## [Reviewer comments · BMJ Open]

ARTICLE DETAILS

TITLE (PROVISIONAL)	Specialized orthotic care to improve functioning in adults with neuromuscular disorders: protocol of a prospective randomized open label blinded end-point study.
AUTHORS	van Duijnhoven, Elza; Koopman, Fieke; Tuijelaars, Jana; Altmann, Viola; Lagrand, Rimke; van Dongen, Johanna; Nollet, Frans; Brehm, Merel-Anne

VERSION 1 – REVIEW

REVIEWER	sarah Tyson University of Manchester UK
REVIEW RETURNED	01-Jun-2020

GENERAL COMMENTS	This is a well written, clear manuscript reporting the protocol for a robust RCT of orthotics for people with neuromuscular disease. Very unusually, there are no areas which I feel need further detail or clarification, It is fine
---

REVIEWER	Aoife Healy Staffordshire University, UK
REVIEW RETURNED	04-Jun-2020

GENERAL COMMENTS	This protocol provides a detailed methodology for a study assessing orthotic care in adults with neuromuscular diseases. As research on the effectiveness and cost-effectiveness of orthotic interventions are limited this research is needed. Minor comments: Add the proposed dates of the study to the manuscript Study population Please clarify if participants who are currently using an orthosis will be included in the study, or will the participants be people receiving their first orthosis? Outcome measures Provide information on if the use of mediation, e.g. pain relievers, be monitored during the study. Gait biomechanics In this section, it is stated that participants will be recorded "walking with the old orthosis if applicable" additional information on this is required. If participants already have an orthosis before the study and they are assigned to the usual care group will they
---

	continue using the current orthosis or will they be prescribed a new one?
--	---

REVIEWER	Simon Lalor Orthotic and Prosthetic Department, The Royal Children's Hospital, Melbourne, Australia
-----------------	--

REVIEW RETURNED	08-Jun-2020
-------------

GENERAL COMMENTS	Firstly a very well put together and thorough study. Study Population: 1) Including patients with peripheral nerve injury. This is a non-progressive condition, in comparison to the other conditions mentioned; CMT, Post-Polio, IBM or MD. By including a group of non-progressive patients to a population of largely progressive diseases, it will skew the results if they all happened to be in one group, as some of the progressive conditions may deteriorate over the 24 week evaluation period. 2) How will you assess and account for the deterioration of the conditions over this 24 week period? Will the patients be reassessed and the orthotic prescription reviewed to ensure it is still appropriate? As you are planning on blinding the patients into the two groups, by not accounting for or matching the conditions, you could introduce a variable that you cannot control, thereby weakening a potentially great study. 3) Assessing both AFOs and KAFOs in one sample: In your introduction you discuss one of the weaknesses of previous studies is the "heterogeneity in the types of leg orthoses" used. This is being repeated in this study by including both AFOs and KAFOs in the groups meaning only generalised recommendations can be produced again. Of greater value would be to focus on one design of orthosis (I would suggest AFOs due to the higher patient use, leading to easier recruitment of patients). I recognise this study is not about purely assessing the function of a specific design of orthosis, but by focusing this study to one device or having groups with the AFO and KAFO users separated, it improves the studies clinical relevance and makes it more specific for the reader's future service planning exercises. 4) If you do split the groups into AFO and KAFO the sample size will need to increase to ensure you have sufficient power for the study. 5) Are participants going to be unilateral users of orthoses or will you include bilateral users? If bilateral users are included, they will need to be singled out/separated or matched in each group as their results will be very different to a unilateral user. This is the difficulty of using the broad umbrella term of "NMD" to define your sample, it is introducing heterogeneity of diagnoses. When you have blinded groups with the varying diagnoses of CMT, Post polio, IBM, MD and peripheral nerve damage, it makes the interpretation of results difficult as each diagnosis will have their own disease specific intricacies and also introduces bilateral and unilateral users into groups together . Methods:
--

	1) Descriptions of the types, components and materials of the orthoses used along with the alignment (shank to vertical and thigh to vertical angles), needs to be thoroughly detailed: see paper: "Do research papers provide enough information on design and material used in ankle foot orthoses for children with cerebral palsy? A systematic review by N. Eddison M. Mulholland and N. Chockalingam 2) Details of footwear used with the orthoses are not provided. Similarly how will the effect of footwear be managed or accounted for over the 24 week evaluation period in terms of shoes wearing out and being replaced. There is the possibility of a patient starting off using appropriate footwear and then swapping half way through to inappropriate footwear and skewing the results: it has been shown in the literature that the construction and sole profile of footwear can have a significant effect on the biomechanics of gait. Outcome Measures: A large number of outcome measures are being used and this must be applauded as there are very few orthotic studies that have adopted a comprehensive approach to research evaluation. 1) Use of oval track for 6MWT: No detail is given of which direction the participants will walk (clockwise or anticlockwise). If the participants are unilateral users then they need to have the orthotic side consistently on the inside or outside of the track (whichever is chosen) so that the results can be compared between patients and groups and eliminate bias. 2) Some of the outcome measures proposed in the study have not been validated for use in orthotic research. It is a limitation of this study in that the results will need to be interpreted carefully as we do not know the MCID, floor/ceiling effects and whether the outcome measures are reliable, repeatable, valid and specific for use in people with a NMD diagnoses using orthoses. 3) Orthotic properties need to be measured at T3: You will need to report the clinical details of the orthosis 24 week mark to ensure the orthoses are fitting well, functioning well, do not display excessive wear, that they are fit for purpose and particularly when using a patient demographic that has progressive diseases, still an appropriate prescription for the clinical presentation 24 weeks after supply. 4) Use of barefoot walking in assessment of gait biomechanics: there will be many people, particularly unilateral KAFO users and bilateral AFO and KAFO users that would be unable to walk unaided and barefoot. How will you account for this in your results?
--	--

VERSION 1 – AUTHOR RESPONSE

Reviewers comments to author

Reviewer 1

This is a well written, clear manuscript reporting the protocol for a robust RCT of orthotics for people with neuromuscular disease. Very unusually, there are no areas which I feel need further detail or clarification. It is fine.

Reviewer 2

This protocol provides a detailed methodology for a study assessing orthotic care in adults with neuromuscular diseases. As research on the effectiveness and cost-effectiveness of orthotic interventions are limited, this research is needed.

Minor comments

Add the proposed dates of the study to the manuscript

Response: As requested, we added the proposed start and end dates of patient recruitment to the manuscript (page 6).

Study population

Please clarify if participants who are currently using an orthosis will be included in the study, or will the participants be people receiving their first orthosis?

Response: Both people who are currently using an orthosis and people receiving their first orthosis are eligible, since for both groups we expect a clinical relevant difference in effectiveness between specialized and usual orthotic care. We clarified this in the study population section (page 6).

Outcome measures

Provide information on if the use of medication, e.g. pain relievers, is monitored during the study.

Response: As part of the economic evaluation, medication use is included in the cost questionnaire and will be monitored at all measurement points. We added a sentence to the outcome measures section for clarity (page 8).

Gait biomechanics

In this section, it is stated that participants will be recorded "walking with the old orthosis if applicable" additional information on this is required. If participants already have an orthosis before the study and they are assigned to the usual care group will they continue using the current orthosis or will they be prescribed a new one?

Response: If participants already use an orthosis at baseline, they will receive a new orthosis prescribed in usual orthotic care when randomized to the control group. To clarify the study procedures for this group, we have added information on the control condition to the usual orthotic care section (page 7).

Reviewer 3

Firstly a very well put together and thorough study.

Including patients with peripheral nerve injury. This is a non-progressive condition, in comparison to the other conditions mentioned; CMT, Post-Polio, IBM or MD. By including a group of non-progressive patients to a population of largely progressive diseases, it will skew the results if they all happened to be in one group, as some of the progressive conditions may deteriorate over the 24 week evaluation period.

Response: While it is true that we include a group of patients with both progressive and non-progressive conditions, we are confident that the decline in muscle strength or physical functioning during the study course of 24 weeks will be negligible based on previous longitudinal studies (Hammarén et al., 2015; Padua et al., 2010; Ter Steeg et al., 2011).

How will you assess and account for the deterioration of the conditions over this 24 week period? Will the patients be reassessed and the orthotic prescription reviewed to ensure it is still appropriate? As you are planning on blinding the patients into the two groups, by not accounting for or matching the

conditions, you could introduce a variable that you cannot control, thereby weakening a potentially great study.

Response: We are confident that deterioration during the study course will be negligible considering the relatively short study period of 24 weeks and the slowly progressive nature of the neuromuscular disorders in our study (also see our response above) .

Review of the orthotic prescriptions will be part of the specialized care and usual orthotic care, as treatment of both groups is embedded in regular care centers. In scope of study purposes, the outcomes will be evaluated at the follow-up measurements, including reassessment of orthotic properties.

Assessing both AFOs and KAFOs in one sample: In your introduction you discuss one of the weaknesses of previous studies is the "heterogeneity in the types of leg orthoses" used. This is being repeated in this study by including both AFOs and KAFOs in the groups meaning only generalised recommendations can be produced again. Of greater value would be to focus on one design of orthosis (I would suggest AFOs due to the higher patient use, leading to easier recruitment of patients). I recognise this study is not about purely assessing the function of a specific design of orthosis, but by focusing this study to one device or having groups with the AFO and KAFO users separated, it improves the studies clinical relevance and makes it more specific for the reader's future service planning exercises.

Response: We agree that assessing AFOs and KAFOs in one sample introduces certain difficulties in generalizing outcomes and with interpreting the results.

Yet, we focus on the (cost-)effectiveness of the orthotic treatment process from a broader perspective, rather than purely assessing the efficacy of certain types of orthoses. Therefore, we feel confident that we are able to address our research question with a combined sample of AFO and KAFO users, as the intervention (specialized orthotic care given according to the Dutch guideline) is developed for this population by means of treatment protocols. Furthermore, participants will be stratified for the severity of muscle weakness (affected proximal versus only distal), to exclude the possibility of inequality of AFO and KAFO users between groups.

If you do split the groups into AFO and KAFO the sample size will need to increase to ensure you have sufficient power for the study.

Response: We have not planned a separate analysis for AFOs and KAFOs, as the current sample size is insufficient for this purpose.

Are participants going to be unilateral users of orthoses or will you include bilateral users? If bilateral users are included, they will need to be singled out/separated or matched in each group as their results will be very different to a unilateral user. This is the difficulty of using the broad umbrella term of "NMD" to define your sample, it is introducing heterogeneity of diagnoses. When you have blinded groups with the varying diagnoses of CMT, Post polio, IBM, MD and peripheral nerve damage, it makes the interpretation of results difficult as each diagnosis will have their own disease specific intricacies and also introduces bilateral and unilateral users into groups together.

Response: We will include both unilateral and bilateral users of leg orthoses. It is true that there is a heterogeneity of diagnoses. Yet, we focus on neuromuscular disorders that all cause non-spastic leg muscle weakness. Although pathologies differ, clinical manifestations that influence orthotic choices, such as severity of muscle weakness, contractures, and joint deformities are similar. Thus, from a treatment perspective, the target population is very much alike, which also became apparent from our previous PROOF-AFO trial, also including both unilateral and bilaterally users of orthoses (Waterval et al., 2017).

Methods

Descriptions of the types, components and materials of the orthoses used along with the alignment (shank to vertical and thigh to vertical angles), needs to be thoroughly detailed: see paper: "Do

research papers provide enough information on design and material used in ankle foot orthoses for children with cerebral palsy? A systematic review by N. Eddison M. Mulholland and N. Chockalingam. Response: We thank the reviewer for the reference. We will take their suggestions into account for the assessment of orthotic properties.

Details of footwear used with the orthoses are not provided. Similarly how will the effect of footwear be managed or accounted for over the 24 week evaluation period in terms of shoes wearing out and being replaced. There is the possibility of a patient starting off using appropriate footwear and then swapping half way through to inappropriate footwear and skewing the results: it has been shown in the literature that the construction and sole profile of footwear can have a significant effect on the biomechanics of gait.

Response: Indeed, the effectiveness of orthoses cannot be evaluated without considering the footwear being used. During the course of the study, the shoes worn (confection shoes if possible or otherwise custom-made shoes), and whether they were replaced, will be documented. If patients use their orthosis in different pairs of shoes, they will be asked to wear the same pair of shoes at all measurement moments. If (newly provided) shoes wear out during the evaluation period, this will affect the outcome of the intervention of which they are part.

Outcome Measures

A large number of outcome measures are being used and this must be applauded as there are very few orthotic studies that have adopted a comprehensive approach to research evaluation.

Use of oval track for 6MWT: No detail is given of which direction the participants will walk (clockwise or anticlockwise). If the participants are unilateral users then they need to have the orthotic side consistently on the inside or outside of the track (whichever is chosen) so that the results can be compared between patients and groups and eliminate bias.

Response: This is an interesting point. We do not have data on the difference in 6MWT performance for unilateral users between walking with the orthosis on the inside or outside of the track. However, we assume that this effect is limited given the large radius of the curves. Considering that the direction of walking during the 6MWT may affect the performance in unilateral users, participants will be asked which direction they prefer and will be allowed to walk clockwise or anticlockwise (similar direction at all measurement points).

Some of the outcome measures proposed in the study have not been validated for use in orthotic research. It is a limitation of this study in that the results will need to be interpreted carefully as we do not know the MCID, floor/ceiling effects and whether the outcome measures are reliable, repeatable, valid and specific for use in people with a NMD diagnoses using orthoses.

Response: For the choice of outcome measures, we followed the recommendations of Brehm et al. (2011) for the assessment of leg orthoses. Although most of the selected outcome measures are either validated for the study population (e.g. walking energy cost, SF-36, FSS), previously used for orthotic research (e.g. Goal Attainment Scaling (GAS) in Phillips et al. (2012)) or both, it is true that we assess some outcome measures that are not validated (e.g. fall rate questionnaire and NRS for stability). This might be a limitation, but it enables the assessment of the patients' functioning from multiple perspectives, covering all ICF levels.

Orthotic properties need to be measured at T3: You will need to report the clinical details of the orthosis 24 week mark to ensure the orthoses are fitting well, functioning well, do not display excessive wear, that they are fit for purpose and particularly when using a patient demographic that has progressive diseases, still an appropriate prescription for the clinical presentation 24 weeks after supply.

Response: Reporting the technical status of the orthosis in terms of fitting, functioning and possible defects at T3 is part of our study procedure and we have added this to the manuscript (Table 1).

4. Use of barefoot walking in assessment of gait biomechanics: there will be many people, particularly unilateral KAFO users and bilateral AFO and KAFO users that would be unable to walk unaided and barefoot. How will you account for this in your results?

Response: Patients at our clinic are routinely assessed for gait biomechanics while walking barefoot. So far, we have measured over 400 patients, and almost all of them, including unilateral and bilateral KAFO users, were able to walk for at least 8 meters without any assistive device, which is sufficient for the assessment of gait kinematics and kinetics. When it is impossible for patients to walk unaided, the gait analysis is performed with minimal assistance and very occasionally barefoot walking is impossible. In case assistance is necessary in such that it affects the test results or barefoot walking is impossible, baseline gait analysis will only include walking with orthosis. We will clearly report this.

References

1. Brehm M, Bus SA, Harlaar J, Nollet F. A candidate core set of outcome measures based on the International Classification of Functioning, Disability and Health for clinical studies on lower limb orthoses. *Prosthet Orthot Int.* 2011;35(3):269-277. doi:10.1177/0309364611413496
2. Hammarén E, Kjellby-Wendt G, Lindberg C. Muscle force, balance and falls in muscular impaired individuals with myotonic dystrophy type 1: a five-year prospective cohort study. *Neuromuscul Disord.* 2015;25(2):141-148. doi:10.1016/j.nmd.2014.11.004
3. Padua L, Pareyson D, Aprile I, et al. Natural history of Charcot-Marie-Tooth 2: 2-year follow-up of muscle strength, walking ability and quality of life. *Neurol Sci.* 2010;31(2):175-178. doi:10.1007/s10072-009-0202-z
4. Phillips MF, Robertson Z, Killen B, White B. A pilot study of a crossover trial with randomized use of ankle-foot orthoses for people with Charcot-Marie-tooth disease. *Clin Rehabil.* 2012;26(6):534-544. doi:10.1177/0269215511426802
5. Tersteeg IM, Koopman FS, Stolwijk-Swüste JM, Beelen A, Nollet F; CARPA Study Group. A 5-year longitudinal study of fatigue in patients with late-onset sequelae of poliomyelitis. *Arch Phys Med Rehabil.* 2011;92(6):899-904. doi:10.1016/j.apmr.2011.01.005
6. Waterval NF, Nollet F, Harlaar J, Brehm MA. Precision orthotics: optimising ankle foot orthoses to improve gait in patients with neuromuscular diseases; protocol of the PROOF-AFO study, a prospective intervention study. *BMJ Open.* 2017;7(2):e013342. Published 2017 Feb 28. doi:10.1136/bmjopen-2016-013342

VERSION 2 – REVIEW

REVIEWER	Simon Lalorlalo Royal Children's Hospital
REVIEW RETURNED	24-Aug-2020

GENERAL COMMENTS	1) Using non progressive and progressive conditions will skew results. CMT shows significant reduction in energy walking cost as measured by 6MWT over a 12 month period. (see article https://pubmed.ncbi.nlm.nih.gov/30926199/) Cannot have CMT and other progressive NMDs along with peripheral nerve palsy in the same cohorts as how do you know the differences you will find in walking ability/activity are due to the orthosis or the NMD progression?
--

	2) How do you account for participants who cannot perform a barefoot walk in the gait analysis? Many people with slowly progressive NMD, particularly KAFO users, will be unable to perform such a task. 3) Including AFOs and KAFOs in the cohort for analysis and comparison is like comparing apples and pears. Both orthoses, both fruit, but both are very different and have different outcomes when applied. If you have all the AFOs randomised in one group and the KAFOs in the other, then your results will be severely biased. The walking energy cost of an AFO user is much less than that of a KAFO user. Given this is one of your primary outcome measures I cannot see how you can be including both in the cohort for analysis. 4) Which direction (clockwise or anticlockwise) of the oval track will the single side orthotic users walk during the 6MWT. Doesn't matter which direction but, needs to be the same for all participants at each test so that results are accurate.
--	---

VERSION 2 – AUTHOR RESPONSE

Reviewers comments to author

Reviewer 3

Using non-progressive and progressive conditions will skew results. CMT shows significant reduction in energy walking cost as measured by 6MWT over a 12-month period. Cannot have CMT and other progressive NMDs along with peripheral nerve palsy in the same cohorts as how do you know the differences you will find in walking ability/activity are due to the orthosis or the NMD progression?

Response: In the suggested article of Pazzaglia et al. (2019), the authors found no change in the mean (SD) walked distance during the 6MWT over a 12-month period in patients with CMT (393.1 (93.0) m versus 393.0 (98.1) m). Furthermore, only a very small decline in the mean walked distance during the 2MWT was found over a 10-year period (-0.6% per year) in a longitudinal study that examined walking capacity in other slowly progressive NMD (Bickerstaffe et al., 2015). Therefore, we are confident that the decline in physical functioning of patients with slowly progressive NMD during our study course of 6 months will be negligible. As such, we believe that we can examine patients with slowly progressive NMDs and peripheral nerve injuries in the same cohort.

How do you account for participants who cannot perform a barefoot walk in the gait analysis? Many people with slowly progressive NMD, particularly KAFO users, will be unable to perform such a task.

Response: For addressing our research question, the relevant conditions that will be assessed during the gait analysis concern walking with shoes only or walking with the old orthosis (at T1) versus walking with the new orthosis (at T2 and T3). We have added a sentence to Table 1 (page 8) in which we describe which data we use for statistical analysis. The barefoot gait analysis condition will only be used for checking the inclusion criteria and will not be used for analysis. Based on

extensive experience, we expect that the majority of participants, also KAFO users will be able to perform a barefoot walk for 8 meters unassisted.

Including AFOs and KAFOs in the cohort for analysis and comparison is like comparing apples and pears. Both orthoses, both fruit, but both are very different and have different outcomes when applied. If you have all the AFOs randomised in one group and the KAFOs in the other, then your results will be severely biased. The walking energy cost of an AFO user is much less than that of a KAFO user. Given this is one of your primary outcome measures I cannot see how you can be including both in the cohort for analysis.

Response: The intervention that will be examined in this study is developed for both AFO and KAFO users, this is why we include both in the cohort. As the groups are different, we will allow for an equal distribution between AFO and KAFO users among treatment groups by stratifying for disease severity, defined as distal leg muscle weakness versus (distal and) proximal leg weakness. This means that patients who only suffer from distal leg weakness (likely mostly AFO users) will be equally allocated to both treatment groups as patients with (additional) proximal weakness (likely mostly KAFO users). By equally distributed groups, we are confident to include both AFO and KAFO users in our study, as their results will be equally weighted in the main analysis.

Which direction (clockwise or anticlockwise) of the oval track will the single side orthotic users walk during the 6MWT. Doesn't matter which direction but, needs to be the same for all participants at each test so that results are accurate.

Response: Considering that the walking direction may affect the performance in unilateral orthosis users, participants will be asked and allowed to walk the direction they prefer, clockwise or anticlockwise. We will document the preferred walking direction and keep this similar at each measurement time point to assure that this will not

influence the results. We have clarified this in the text (Page 8).

References

1. Pazzaglia, C., Padua, L., Pareyson, D., Schenone, A., Aiello, A., Fabrizi, G. M., ... & Gemignani, F. (2019). Are novel outcome measures for Charcot–Marie–Tooth disease sensitive to change? The 6-minute walk test and StepWatch™ Activity Monitor in a 12-month longitudinal study. *Neuromuscular Disorders*, 29(4), 310-316.
2. Bickerstaffe, A., Beelen, A., & Nollet, F. (2015). Change in physical mobility over 10 years in post-polio syndrome. *Neuromuscular Disorders*, 25(3), 225-230.